# Long-Term Performance of Nanomodified Coated Concrete Structures under Hostile Marine Climate Conditions

**DOI:** 10.3390/nano11040869

**Published:** 2021-03-29

**Authors:** Adrián Esteban-Arranz, Ana Raquel de la Osa, Wendy Eunice García-Lorefice, Javier Sacristan, Luz Sánchez-Silva

**Affiliations:** 1Department of Chemical Engineering, University of Castilla-La Mancha, Avenida Camilo José Cela, 12, 13071 Ciudad Real, Spain; Adrian.Esteban@uclm.es (A.E.-A.); AnaRaquel.Osa@uclm.es (A.R.d.l.O.); Wendy.Garcia@uclm.es (W.E.G.-L.); 2ACCIONA Technological Centre, Alcobendas, 28108 Madrid, Spain; sacristan.javier@es.sika.com

**Keywords:** epoxy resins, thermal stability, curing degree, water permeability, adherence, abrasion resistance

## Abstract

Epoxy resin coatings are commonly used to protect concrete structures due to their excellent chemical corrosion resistance and strong adhesion capacity. However, these coatings are susceptible to damage by surface abrasion and long-term contact with marine climate conditions, deteriorating their appearance and performance. This study aims to optimize the performance of cement-based epoxy resin coatings, bisphenol-A and polyol, in aggressive environments by functionalizing the selected systems with different nanoparticles such as activated carbon, surface modified nanoclay, silica and zinc oxide. Nanomodified coatings were applied to concrete specimens and subjected to three weeks in a spray salt chamber and three weeks in a QUV chamber. They were found to present improved thermal resistance and curing degree after the weathering test. Their water permeability, adhesion, and abrasion resistance properties were evaluated before and after this test. The results showed that the nature of the nanocomposites determined their water permeability; the bare resin presented the worst result. Additionally, nanomodified composites with activated carbon and silica showed the best adherence and abrasion resistance properties, due to the effect of this aging test on their thermal stability and curing degree.

## 1. Introduction

Concrete is extensively used in marine and coastal defense frames such as tunnels, quay walls, bridges, jetties, breakwaters, or sea walls, whose structural design is supposed to last more than 50 years. Thus, the durability and structural reliability of concrete structures are critical factors that require special attention, since they depend on a combination of environmental, design factors, and construction practices. In any case, concrete structures located in marine conditions suffer from continuous exposure to seawater, either in a direct or indirect way, which leads to a wide range of deterioration scenarios, e.g., functional, aesthetic, or structural problems.

Parameters such as temperature, pH, the type of cement, the number of harmful substances, and the permeability of concrete are found to be key for the chemical attack. Generally, seawater contains 3.5 wt.% of dissolved salts, mainly in form of Na^+^ and Cl^−^ (11,000 and 20,000 mg·L^−1^, respectively) and Mg^2+^, Ca^2+^, K^+^, SO_4_^2−^ to a lesser extent, which enters the pores of concrete and chemically reacts with its components causing damage to the structure. Among the harmful substances present in a pore solution, magnesium sulfate may react with calcium hydroxide, resulting in calcium sulfate and magnesium hydroxide precipitation, or with hydrated calcium aluminate, forming calcium sulfo-aluminate, which are the main cause of expansion and leaching actions on concrete, affecting the strength of the structure. Chloride ions significantly contribute to the reduction of service life, but deterioration is also linked to the loss of part of the lime content of concrete due to leaching. Nevertheless, concrete structures are also subjected to contamination by gases, such as O_2_, N_2_, and CO_2_, which are dissolved in seawater and enter the pores, lowering the pH to below 7.5 (compared to the nominal pH of 8.2). Consequently, these harmful effects induce the oxidation and degradation of reinforced concrete bars.

On the other hand, apart from chemical deterioration processes, physical degradation also takes place. Although the most damaged area of a concrete structure exposed to marine conditions is known to be above the high water mark due to the continuous direct impact of waves on that area, erosion and abrasion by direct contact of sand and silt with the concrete surface, especially in shallow waters, must be taken into consideration. As the outer paste of concrete wears, the inner aggregates are exposed to further abrasion, causing extra degradation. In addition, turbulent movements of water may generate vapor cavities that can implode when subjected to high pressure, inducing an intense shock wave, resulting in the delamination of the structure. Also, differences of temperature can cause different expansion coefficients for the aggregates and the cement matrix, leading to tensile stresses with crack risk. Furthermore, frost and de-icing in concrete pores provoke expansion, disruption, and a loss of durability.

Therefore, the development of a method to mitigate degradation under highly aggressive environments is still a matter of interest. In this sense, different strategies can be followed to improve concrete durability:i.high pressure steam cured concrete elements,ii.the use of a suitable concrete specification, cover thickness, and compaction of construction joints;iii.the adoption of alternative materials to steel as reinforcement bars;iv.the use of concrete protective coatings; orv.cathodic barrier techniques.

Considering the cost of replacement and repair for damaged concrete structures in coastal zones, the use of coatings may provide a cost-effective method to significantly extend service life. In this context, epoxy coatings seem to be a suitable alternative to increase the level of protection due to their feasibility for use with all kinds of concrete, excellent adhesion, environmentally friendly properties, and durable waterproofing [1]. Although this kind of resin supplies chemical resistance, working as a barrier to acidic gases, moisture, and other contaminants [2], some concerns have been raised regarding weak mechanical properties. To solve this drawback, epoxy resins require the support of inorganic materials, fibers, or synthetic polymers through nanoscale blending [3,4]. This way, interest in advanced epoxy nanocomposites has recently increased due to their singular physicochemical properties, as a result of the combination of those from the epoxy-based material and the special characteristics of the nanoparticles (NPs), which could also fill the holes and defects of epoxy coatings, preventing cracking.

In order to enhance the mechanical performance and introduce new functionalities different nanostructure materials such as iron-based nanoparticles [5], silica [1,6,7,8], alumina [9], nanoclay [10,11,12], zinc oxide [1,13,14,15], and carbonaceous materials [16,17,18,19,20,21] (e.g., activated carbon, carbon nanofibers, graphene, or carbon nanotubes) have been used to prepare epoxy nanocomposites [22]. The incorporation of ZnO particle filler to transparent epoxies was found to significantly enhance UV light resistance [14] and thermal stability. SiO_2_ supplies good adhesive [23] and mechanical properties, along with excellent thermal stability and corrosion and carbonation resistance. However, their hydrophilic behavior may confer a lower impermeability to the coating. Carbonaceous fillers have been found to greatly enhance mechanical properties, probably due to their high specific surface area and good compatibility with most polymers (enhanced nanofiller–matrix adhesion). The use of modified nanoclay increases the hydrophobicity of the coating and provides uniform NP distribution within the epoxy matrix. The improvement of the nanocomposite properties is significantly influenced by the NP concentration (dosage limit), dispersion, and adhesion quality in the polymer matrix [24]. Thus, epoxy nanocomposites with advanced multifunctionalities still need to be developed and adjusted to fit protection demands.

Therefore, this work aims to discuss the results obtained for the application of three different commercial epoxy-based coatings, i.e., cement epoxy resin (SE), bisphenol-a epoxy resin (S) and polyol epoxy resin (M), on the characteristics of a standard concrete specimen, and the subsequent selection of the most suitable one to prevent the deterioration of marine constructions. The influence of the incorporation of a certain load of nanoparticles to reinforce the selected epoxy resin-based coating was also explored. In that respect, NP loading and way of incorporation were firstly optimized. Then, the physicochemical properties of the as-prepared nanocomposites related to the addition of NPs of different nature (i.e., zinc oxide, silica, charcoal activated powder and a surface modified nanoclay with 35–45 wt.% dimethyl dialkyl (C14-18) amine) were evaluated. NP wettability (in terms of the hydrophobic/hydrophilic behavior), as well as the surface chemistry and thermal stability of the different composites, were analyzed and compared to those obtained by the NP-free coated specimen. In a last stage, each coated concrete test specimen was subjected to a series of test: i) water permeability (absorption), ii) adhesion (by means of pull off), and iii) abrasion (Taber), before and after an accelerated weathering test simulating a marine environment in a salt chamber, which consisted of a three-week wetting cycle (35 °C), followed by another three weeks in a UV chamber (interchanging 4 h UV 340 nm radiation at 60 °C and 4 h of condensation at 50 °C).

## 2. Materials and Methods

### 2.1. Materials and Concrete Specimen Preparation

In this study, concretes with a nominal 28-day strength of 35 MPa and 110 mm slump were used as the base material. Different dosages of ordinary Portland cement, coarse aggregates with a maximum size of 20 mm and a density of 2.6 g·cm^−3^, and sand as fine aggregates were included in the mixtures. The main mixing equipment was the vertical axis forced concrete mixer. Coupons of different dimensions (100 × 50 × 30, 127 × 76 × 30 and 100 mm × 100 mm × 30 mm) were cured for 14 days before being removed from the formwork after pouring. Ambient conditions during casting and curing did not vary significantly. Then, the concrete structures were prepared to improve their performance. Concrete structures were coated with three different commercial coatings:i.a cement-based epoxy resin (SE);ii.a bisphenol-A based epoxy resin; andiii.a polyol-based epoxy resin.

According to the recommendations of the supplier, a primer coating for S and M was needed in order to improve the adhesion and prevent the appearance of pinholes or bubbles in the hardened coating. More details of these coatings are given in Appendix A.

Four types of nanoparticles were used in this study to enhance the physicochemical properties of the resins: charcoal activated powder (AC), surface modified nanoclay with 35–45 wt.% dimethyl dialkyl (C14-18) amine (SMNC), silicon dioxide with 99.5 wt.% purity (SiO_2_) and zinc oxide with 99 wt.% purity (ZnO). AC and ZnO were purchased from PanReac (Barcelona, Spain), and NCSM and SiO_2_ were purchased from Sigma Aldrich (Madrid, Spain).

### 2.2. Optimization and Characterization of the Coating Methodology

The three commercial coatings (SE, S, M) previously described were considered in this optimization. To impregnate them, four film deposition methodologies were studied:i.spatula printing;ii.paintbrush impregnation;iii.paintroller impregnation; andiv.dip-coating methodology.

In the case of the spatula printing (i), SE coating was applied according to the recommendation of the supplier (Appendix A). First, Part A and Part B had to be mixed vigorously. Then, Part C was gradually incorporated into the mixture (A + B) while stirring. After obtaining a uniform mixture, the coating was applied and spread among the surface of the concrete structures. The waiting time (at room temperature) to ensure a proper drying of the coating was 24 h. In the case of S and M resins, firstly, Part A and Part B had to be vigorously mixed until a homogeneous mixture was obtained. Then, the coating was applied to the surface of concrete structures with a brush or roller. In these cases, the drying time was also 24 h at room temperature. These resins can be also spread on the surface of the concrete by a dip-coating procedure. This approach consisted of immersing the concrete structures for a specific time into the solution of the coating resin (A + B homogeneous mixture) to ensure that the substrate was fully infiltrated. Then, the structures were slowly removed from the solution and allowed to dry at room temperature. The type of coating, the immersion time, and the number of immersions were chosen to further optimize the coating methodology. For the latter, the first resin was left to dry for one day at room temperature before the second dip coating was applied. The nomenclature followed hereafter in this work is Ct-N, where C is the type of coating (SE, S or M), t is the immersion time (in minutes), and N is the number of coating immersions. Coated concrete structures were then exposed to an artificial accelerated weathering test based on a slight modification of the ASTM D 5894 [25] that run for six weeks. In this study, concrete structures were placed into a salt fog chamber for three weeks before being moved to a fluorescent UV/condensation chamber for another three weeks. The salt fog experiments were carried out using Salt Spray Equipment (SSE) from Dycometal S.L at 35 °C with a sodium chloride solution (5 wt.% purity purchased from PanReac, Barcelona, Spain) flow of 2 mL·h^−1^. The fluorescent UV/condensation chamber was a QUV Accelerated Weathering Tester, from Q-Lab Corporation. To provide a more realistic marine weather exposure regime, the accelerated aging test consisted of cyclic exposure of UVA-340 light at 0.65 W·m^−2^ at 60 °C for 4 h and condensing humidity at 50 °C in the dark for 4 h. The radiation intensity was calibrated by a radiometer every 150 h. Water permeability tests of the coated concrete structures (127 mm × 76 mm × 30 mm) were carried out following the UNE EN 1062 [26]. The test consisted of three cycles of immersing the concretes in distilled water for 24 h followed by 24 h of drying in an oven at 50 °C. This test was repeated twice to check the reproducibility of the results. The adhesion strength of the coatings to the concrete structures (100 mm × 50 mm × 30 mm) was evaluated with the pull off test. An adhesion tester Elcometer 506 (Elcometer^®^) (Manchester, England) was applied to the front face of the concrete structures, following the ASTM D 4541 [27]. A pull off test was performed three times, with the final adhesion strength value being the average of those measurements. The diameter of the dolly used for the pull off test was 20 mm; the adhesive was purchased from Araldit Standard Professional (Basel, Switzerland). Accelerated abrasion resistance tests were carried out using the Taber abraser method at room temperature (25 °C) and 45–50% humidity by means of a Taber Rotary Abraser model 5135, from Taber^®^ Industries (North Tonawanda, NY, USA). Each specimen, with dimensions of 100 mm × 100 mm × 30 mm, was loaded for a total of 6000 cycles under an applied load of 1000 g in accordance with ASTM D 4060 [28], with two CS-17 abrading wheels. The abrasion resistance values were calculated through the mass loss in a specified number of abrasion cycles with the selected loads.

### 2.3. Characterization of Materials

Contact angle micrographs were acquired by using an Attention Theta Optical Tensiometer from Biolin Scientific’s (Madrid, Spain) with the OneAttension software (Washington, DC, VA, USA) and accuracy of ±0.1° precision. Contact angle values were recorded instantaneously after the water droplet touched the surface of the materials to minimize evaporation errors. Analyses were performed in three equidistant points of the sample, with the final contact angle values being the average of three measurements. Micrographs of the different NPs were acquired using a GeminiSEM 500 High Resolution Scanning Electron Microscope (HRSEM) from ZEISS brand (Oberkochen, Germany). The surface chemistry of the different materials was elucidated using a Spectrum Two Fourier transform infrared (FTIR) spectrometer, from PerkinElmer Inc. (Madrid, Spain), with a zinc selenide (ZnSe) crystal. Spectra ranged from 4000 to 500 cm^−1^, with a resolution of 4 cm^−1^, and 100 scans per sample were recorded. Thermogravimetric analyses (TGA) were evaluated using a Mettler Toledo TGA/DSC 1 STARe under nitrogen and air atmospheres (90 NmL·min^−1^ flow rate) at a heating rate of 10 °C·min^−1^ from 25 °C to 1000 °C. Differential scanning calorimetry (DSC) experiments were performed using a Mettler Toledo DSC2 under nitrogen atmosphere with a gas flow of 50 NmL·min^−1^ and a heating rate of 10 °C·min^−1^ in the temperature range from 25 to 300 °C. Dispersion stability analyses of the NPs in Part B of the resin were determined by the Stability Index (TSI) provided by the TurbiscanTM Lab Expert stability analyzer, from Formulation Company, Toulouse, France. Samples were prepared in cylindrical vials with 1 wt.% of each type of NP. They were dispersed for 10 min in 20 mL of Part B using an ultrasonic bath (215/860 W, 50/60 Hz, Bandelin Sonorex Digiplus) and rapidly placed into the analyzer. Stability experiments were performed at 30 °C for 24 h and data were collected every 30 s for the first 1.5 h and every 30 min for the rest of the experiment. The thickness of the resulting coatings was evaluated using a stereoscopic microscope optic loop with 7.5× zoom and 115 mm working distance, manufactured by Nikon (Amstelveen, The Netherlands). Density measurements were carried out using a pycnometer (Ultrapyc 1200-e, Boynton Beach, FL, USA) with a medium cell size. Adsorption–desorption of nitrogen at −196 °C analyses were recorded in a Gemini VII (Micromeritics, Norcross, GA, USA). Samples were degassed at 120 °C overnight prior their measurement. Surface area values (S_BET_) were calculated using the Brunauer Emmett Teller equation.

### 2.4. Optimization of the NP Incorporation into the Coatings

AC, SMNC, SiO_2_, and ZnO nanoparticles were selected for this study. Two different NP concentrations were considered, i.e., 1 and 5 wt.%, to be dispersed in Part B of the resin. First, 5 wt.% of nanoparticles were dispersed in Part B (hardener) by sonication (400 W, 50/60 Hz, UP400S Hielscher) for 10 min at room temperature, which was then vigorously mixed with Part A. After that, concrete structures were immersed in the resulting solution. The same procedure was carried out with 1 wt.% of NPs. In addition, this NP loading was dispersed in Part B via an ultrasonic bath (215/860 W, 50/60 Hz, Bandelin Sonorex Digiplus) for 10 min.

## 3. Results and Discussion

### 3.1. Characterization and Optimization of the Coating Methodology

In Appendix A, the different coating methodologies with their respective concrete specimens are depicted. Macroscopically, a more homogeneous and dispersed coating was obtained with the dip-coating approach, so further results were obtained using this methodology. In spite of the fact that the dip coating methodology displayed the most homogeneous and best-dispersed coating at the lab scale, we believe that spray coating would be a more appropriate approach for industrial applications.

Water permeability and pull-off tests were carried out for the different coated concrete structures. For comparison purposes, these tests were also run with a raw concrete specimen as a blank. Figure 1 summarizes the results corresponding to:i.type of resin;ii.coating time;iii.and number of the coating immersions;

As shown in Figure 1a, the water permeability test showed an improvement with the application of the S resin in comparison to the other resins and the blank specimen. In addition, a pull-off test proved that better adhesion results were obtained with this resin (8.5 MPa), followed by the M resin (7.2 MPa). The cement coated concrete (SE) displayed half of the adhesion strength (3 MPa) in comparison to the other resins. Similar results were observed elsewhere [29]. Therefore, subsequent optimization studies were only performed with S and M resins. From Figure 1b, it can be concluded that longer immersion times had a negative effect on the water permeability and adherence properties for both resins. In this sense, a greater amount of water was absorbed, and lower pressures were needed to detach the different coatings from the concretes, especially for M resin (6.7 MPa). Thus, S resin was selected for the optimization of immersions repetitions. In this case, similar water permeability and adherence results to those of S10-1 were obtained.

To understand these results, the thickness of both S and M coatings and their contact angle results were measured and are summarized in Figure 2. Figure 2a shows that thinner coatings were obtained with S resin in comparison to M resin. These results suggest that thinner coatings could promote stronger interactions between the concrete and epoxy resin, resulting in better adherence behavior of the coating. Additionally, M resin displayed a more hydrophilic nature in comparison to S resin. This fact helps to explain its worse water permeability results. Otherwise, thicker and more hydrophilic coatings were found after longer times of immersions with both epoxy resins (S20-1 and M20-1), as shown in Figure 2a,b, which caused worse adherence and water permeability results. Finally, similar thickness and contact angle values to those of S10-1 were obtained after a second immersion of the S resin (S10-2). Therefore, immersing the concrete only one time in the S epoxy resin for ten minutes was established as the optimal coating condition.

### 3.2. Characterization of Commercial Nanoparticles

The physical and textural properties of the different NPs are summarized in Appendix A. ZnO displayed a greater density value in comparison to the other NPs, following the trend: ZnO (6.1 g·cm^−3^) > SiO_2_ (2.7 g·cm^−3^) > SMNC (1.7 g·cm^−3^) > AC (1.1 g·cm^−3^). The adsorption–desorption of nitrogen at −196 °C results evidenced that AC and SiO_2_ materials possessed higher surface area values, i.e., 954.8 and 533.3 m^2^·g^−1^, respectively, in comparison to SMNC (11.8 m^2^·g^−1^) and ZnO (5.4 m^2^·g^−1^). In Appendix A HRSEM micrographs and contact angle analyses of the different nanoparticles are shown, respectively. AC presented a highly porous texture, which is usually associated with a high specific surface area, as evidenced in Appendix A and reported elsewhere [30]. In this case, its highly porous structure could be the reason of a contact angle value lower than 90°, despite being a carbon-based material. In the case of SMNC, the surface modification with amines of the starting nanoclay led to flake-shaped particles with hydrophobic properties [31], as the contact angle value was close to 125°. Otherwise, SiO_2_ was composed of spherical particles, while ZnO displayed a hexagonal structure. The contact angle values obtained for these nanoparticles were lower than 90°, displaying a hydrophilic nature. Next, ATR-FTIR experiments were carried out to define the surface chemistry of the different NPs. Their spectra are shown in Appendix A. AC showed less prominent vibration modes in comparison to the other nanoparticles due to its black appearance [32]. In the case of the SMNC spectrum, the bands near 2963, 2922 and 2854 cm^−1^ were assigned to the stretching vibration of aliphatic groups (C-H) in organic matter [33]. The bands at about 1450 cm^−1^ could be assigned to aliphatic C-H deformation vibrations due to the presence of polymethylene [33,34]. The presence of allophane in SMNC was also confirmed due to the bands near 670 and 550 cm^−1^. The amine groups from the surface modification were detected at 3620 cm^−1^ and 917 cm^−1^. Si-O stretching vibration of orthosilicate anions and Si-O-Al groups were shown at about 1004 cm^−1^ [35,36]. For SiO_2_, the absorption bands assigned to Si-O stretching vibrations appeared at 1069 cm^−1^, and Si-O bending vibrations at 814 cm^−1^ were also identified [37,38,39]. The FTIR spectrum of ZnO presented an absorption band at around 500 cm^−1^ that clearly demonstrated the presence of stretching Zn-O bonds [40].

### 3.3. Optimization of Commercial Nanoparticle Incorporation into the Coating

Previous studies have investigated the effect of the incorporation of different NP concentrations into epoxy anticorrosive coatings. Bagherzadeh et al. [41] studied the properties of these coatings using SiO_2_ NPs in the range of 1 to 5 wt.%, and Tomic et al. used a surface modified nanoclay in the range of 1 to 10 wt.% [10]. The best results were obtained with less than 5 wt.%. Therefore, in this work, the addition of 1 and 5 wt.% NPs was considered to optimize their incorporation into the coating (Appendix A). SiO_2_ was selected as an example NP for this optimization process. First, 5 wt.% of SiO_2_ was dispersed into Part B by sonication. In a similar way as in the studies by Bagherzadeh and Tomic, the incorporation of 5 wt.% of SiO_2_ into the coating presented visual agglomerations, resulting in a nonhomogeneous coating (first trial). In order to enhance the dispersion and diminish NP agglomeration, 1 wt.% of SiO_2_ was dispersed in Part B of the coating using the same procedure. In this case, there was no evidence of visual agglomeration of NPs (second trial). However, a nonuniformed coating was still observed. According to previous studies [42,43], high energetic cavitation bubbles could be generated during sonication, leading to scissions of the polymeric chains of the coatings. Hence, coatings with 1 wt.% SiO_2_ were prepared using an ultrasonic bath as the dispersion technique to avoid the occurrence of such scissions. As a result, the obtained coated concrete structures presented a uniform coating without NP agglomeration (third trial). Therefore, 1 wt.% of NPs dispersed by ultrasonic bath was chosen as the optimal methodology for the incorporation of the different NPs into the coatings. As shown in Appendix A, homogeneous and well dispersed coated concretes were obtained with all the NPs-S10-1.

### 3.4. Characterization of the Incorporation of NPs into the Part B of the Resin

To define the impact of NP incorporation into the coating, a characterization of the NPs in Part B was carried out. The stability of the different NPs in Part B of the epoxy resin was evaluated (Figure 3a) by Turbiscan experiments. Part B of the epoxy resin was also tested as a blank experiment. This procedure has previously been described to define the stability of functionalized nanomaterials in different solvents [44]. Based on the classification presented by Dai et al., three stability regions could be found, depending on the Turbiscan Stability Index (TSI) values:i.well dispersed materials (TSI < 5);ii.part of the material was deposited at the bottom of the flask and could be easily re-dispersed (5 < TSI < 20);iii.the material was mostly sedimented (TSI > 20) [45].

At 10 min of dispersion, all NPs displayed a TSI value lower than 5, so it could be considered that these NPs were well dispersed in Part B of the resin for the selected time. The TSI values for these NPs at 24 h were also determined. In this case, the best results were obtained with AC (TSI = 0.3) in comparison to SMNC (TSI = 12.2) and SiO_2_ (TSI = 10.5), ZnO (TSI = 16.5). This fact could be explained by its high porosity, as evidenced by HRSEM (Appendix A), and surface area (Appendix A). AC is likely able to incorporate the Part B of the epoxy resin in its pores, leading to better a dispersion [46].

To determine the thermal effect of NP incorporation into Part B of the resin, TGA experiments in nitrogen atmosphere were carried out; the results are shown in Figure 3b. Part B of the resin displayed two principal weight losses at T_1_ = 148 °C and T_2_ = 219 °C. The NP incorporation showed a positive effect on the thermal stability of the NPs-Part B composites. This effect was more pronounced after the incorporation of hydrophilic NPs, SiO_2_ (T_1_ = 193 °C and T_2_ = 251 °C) and ZnO (T_1_ = 191 °C and T_2_ = 251 °C) in comparison to hydrophobic NPs, SMNC (T_1_ = 175 °C and T_2_ = 243 °C) and AC (T_1_ = 183 °C and T _2_= 246 °C).

### 3.5. Characterization of the Coatings before and after the Weathering Test

The results from the characterization of the different S10-1 and NPs-S10-1 coatings (Part A + Part B) before and after the weathering test are summarized in Table 1. Contact angle and optical loop micrographs of the different NPs-S10-1 composites before and after the weathering tests are shown in Appendix A, respectively. NPs-S10-1 composites could be classified as hydrophilic materials in both cases (before and after the weathering test), so a drastic change of the nature of the composites was not detected. Before the weathering test, the incorporation of AC and SMNC nanoparticles into the resin changed the nature of these composites into less hydrophilic materials, obtaining a contact angle of θ_AC-S10-1_= 77.8° and θ_SMNC-S10-1_= 88.7°, respectively, in comparison to θ_S10-1_= 71.6°. The integration of SiO_2_ and ZnO promoted more hydrophilic composites, θ_SiO2-S10-1_= 63.8° and θ_ZnO-S10-1_= 59.7°, respectively. After the weathering test, all composites displayed greater contact angle values in comparison to the S10-1 resin, i.e., ZnO-S10-1, which presented the least hydrophilic nature. As mentioned in research published elsewhere, ZnO could promote photocatalytic reactions in the resin upon exposure to UV-vis light that could make the composite less porous and more hydrophobic [47,48], indicating promising properties for future water permeability applications.

Regarding the optical loop results before the weathering test (Table 1), the incorporation of more hydrophobic nanoparticles led to thinner coatings in comparison to those from more hydrophilic nanoparticles. After the weathering test, a decrease in the thickness of the coatings impregnated with more hydrophilic nanoparticles was observed, especially for ZnO-S10-1 (84.1 µm). This finding was probably due to the reduction of porosity, as previously explained. DSC experiments were performed to study the transition temperature of the pristine resin S10-1 and its derived composites before and after the weathering test (Figure 4). Their glass transition temperature (Tg_1_) values are summarized in Table 1. After the incorporation of the different nanoparticles, a second phase at T~145 °C was produced, due to interactions between the polymer chains of the epoxy resin and the surface charge of the NPs [49]. This effect was more pronounced for AC-S10-1 and SiO_2_-S10-1. Before the weathering test, the incorporation of different NPs promoted a more ordered composite and higher Tg_1_ values (Figure 4a), following the order: Tg_SiO2-S10-1_ (58.6 °C) > Tg_SMNC-S10-1_ (57.6 °C) > Tg_AC-S10-1_ (56.9 °C) > Tg_ZnO-S10-1_ (56.8 °C) > Tg_S10-1_. This increment in the Tg could be attributed to the limited mobility of the epoxy chains due to the intense interactions between the NPs and the epoxy resin [50,51]. In this sense, the addition of 1 wt.% of SiO_2_ led to a maximum Tg ~ 59 °C. Similar results were obtained by Rawaiz Khan et al. with the incorporation of 2 wt.% of SiO_2_ into an epoxy resin [50]. After the weathering test (Figure 4b and Table 1b), less ordered composites were obtained. Moreover, Tg_1_ was shifted to higher temperature values, especially for AC-S10-1 and SiO_2_-S10-1 composites, which could be associated with better thermal stability properties and curing degree of the coatings.

To further investigate the surface interactions between the S10-1 epoxy resin and the different NPs, FTIR-ATR experiments were carried out; the results are shown in Appendix A. In the case of Part A of the resin, two characteristic bands from the oxirane ring were detected (Appendix A). The first one, located at 3057 cm^−1^, was attributed to the C-H in the methylene group of the epoxy group. The second one at 916 cm^−1^ was characteristic of C-O deformation from the oxirane group [52]. After mixing Part A with Part B, the curing process took place, consequently producing S10-1. In this case, a broad band centered at 3400 cm^−1^, correlating with the O-H stretching mode of hydroxyl groups produced after the ring opening reaction, was detected [53]. In this sense, most of the contribution from the oxirane group present in Part A of the resin was eliminated, proving its high curing degree (Appendix A). Additionally, from the FTIR-ATR spectra shown in Appendix A (1000 cm^−1^ to 850 cm^−1^), it can be observed that the incorporation of the NPs improved the curing process of the S10-1, in agreement with what was observed using DSC. The thermal stability of the different composites before and after the weathering test under nitrogen and air atmospheres was evaluated by TGA. The results of the bare coating, S10-1, and its composites, NPs-S10-1, are depicted in Appendix A and summarized in Table 1. In all cases, three different weight losses were detected. The first started at T = 135–150 °C and corresponded to the homolytic scission of the chemical bonds. The second started at T = 330–375 °C and was related to the removal of water molecules and the subsequent formation of double bonds, leading to the breakdown of the epoxy network [54,55]. The final weight loss, which started at T = 510–530 °C, was attributed to the formation of radicals during the degradation of the resin after the isomerization, intramolecular cyclisation, and chain transfer reactions [56]. It was previously established that the temperature at 5% weight loss (T_5%_) corresponds to the initial thermal degradation temperature. T_5%_ values are summarized in Table 1. Before the weathering test, more hydrophilic NPs-S10-1 composites displayed higher T_5%_ values in comparison to neat epoxy resin in nitrogen (Table 1a). Additionally, the different temperature values of NPs-S10-1 at 40% weight loss (T_40%_) were recorded (Table 1a).

Regarding nitrogen atmosphere results, the incorporation of nanoparticles into the coating promoted higher temperatures than the neat epoxy resin, especially for SMNC-S10-1. However, the results obtained in air showed that lower temperatures were found with NPs-S10-1, proving that the experimental atmosphere has an important effect on the degradation of the resin. Attending to results obtained after the weathering test (Table 1b), higher T_5%_ and T_40%_ values were found in comparison to those acquired for the composites before the weathering test (Table 1a). Therefore, this aging treatment (fog/UV-vis) helped to make these composites more stable in both atmospheres. The results in nitrogen atmosphere showed that the nature of the NPs dictated their T_5%_ behavior, since less hydrophilic composites, i.e., AC-S10-1, and SMNC-S10-1, displayed lower temperature values (T_5% =_ 195.2 °C and 191.3 °C) than the neat epoxy resin (T_5%_ = 201.7 °C). Otherwise, T_40%_ results showed that these composites were the most thermally stable at higher temperature, i.e., T_40%_ = 414.5 °C and T_40%_ = 417.5 °C, respectively. In the case of air atmosphere, AC-S10-1 was the only composite which promoted higher temperatures (T_5%_ = 177.1 °C) than the neat epoxy resin (T_5%_ = 175.5 °C). The same trend was obtained with this composite for T_40%_ results, meaning that this material yielded the best experiment results, T_40%_ = 404.8 °C.

Analyzing in detail the different weight losses related to each composite during the heating process, the impact of NP incorporation at every stage of these thermal processes can be established. As previously explained, each weight loss was attributed to a different degradation reaction. All results and final residues at 1000 °C are compiled in Table 1. The type of atmosphere presented an important influence on the process, since the weight loss produced by the homolytic scissions of the chemical bonds from the resin (stage 1) increased under air atmosphere. Otherwise, the weight loss from the breakdown of the epoxy network (stage 2) was enhanced under nitrogen atmosphere. In this case, the incorporation of NPs into the epoxy resin reduced this reaction. Additionally, it is worth mentioning that after the weathering test, an important decrease in the Stage 1 weight loss for all composites under both atmospheres was detected, yielding greater residue values at 1000 °C.

### 3.6. Results of Water Permeability, Pull-off, and Taber Test Before and After Weathering Test

The different coated concretes were subjected to an accelerated weathering test (fog/Uv-vis) to simulate the effect of hostile climate conditions. In this sense, the water permeability, adherence capacity and abrasion resistance of the different coatings were evaluated; the results are depicted in Figure 5. Before the weathering test, water permeability experiments showed that less hydrophilic materials, as determined in the contact angle experiments, AC-S10-1, and SMNC-S10-1, yielded better results in comparison to the neat epoxy resin (Figure 5a). After the weathering test, all NPs-S10-1 composites displayed better water permeability in comparison to the starting epoxy resin, with ZnO-S10-1 presenting the best results. This improved property can be attributed to the least hydrophilic character of this coating after the incorporation of ZnO, which likely reduced the porosity of the composite, providing less free space for the water to permeate into the pores of the coating [47,48]. The results obtained from the pull-off test before the weathering test (Figure 5b) showed that although all composites except for SiO_2_-S10-1 promoted better adherence properties than S10-1, the best results were achieved with the least hydrophilic composites, i.e., AC-S10-1, and SMNC-S10-1. However, the latter was the only one that showed worse adherence than S10-1 after the weathering test. The best results were therefore achieved with AC-S10-1 and SiO_2_-S10-1 composites, with the latter showing superior performance. This behavior could be explained by taking into account the DSC results (Figure 4b), where higher temperatures acquired with these nanocomposites after UV exposure were due to their improved curing reactions and promotion of the postcrosslinking effect of their polymeric network structures [57].

Additionally, the Tg for both nanocomposites evidenced stronger interaction between the polymer network and these nanoparticles. Finally, abrasion resistances results, before and after the weathering test, are shown in Figure 5c. It is important to note that after the aging test, all composites displayed better abrasion results. As previously demonstrated by TGA experiments, the fog/UV-vis cycle treatment improved the thermal properties of all composites in air and nitrogen atmospheres (Appendix A).

However, AC-S10-1 and SiO_2_-S10-1 were established as the only composites that promoted better Taber wear index values in comparison to S10-1 after the 6000 cycle abrasion test. These findings were in accordance with our previous pull-off results, since stronger adherence properties were developed with these materials, as evidenced by DSC experiments (Figure 4b) [58]. The great mass loss found with ZnO-S10-1 composite could be due to the capacity of the metal oxide to react with UV light and produce OH^•^ free radicals from water and oxygen [59]. These radicals are considered high oxidant species that could attack and degrade the polymeric network [60]. Evidence of this reaction would be the thinner coating obtained by the optical loop (Table 1). Therefore, it was shown that this accelerated weathering test can help in the development of advantageous coating properties for future research of concrete applications.

## 4. Conclusions

In this work a coating methodology of concrete based on different epoxy resins was optimized. Accordingly, the immersion of a concrete specimen for 10 min in bisphenol-A based epoxy resin was established as the optimal coating methodology. The dispersion of 1 wt.% of nanoparticles into Part B of this resin for 10 min of ultrasonic bath resulted in homogeneous and well dispersed coatings, with AC presenting the best dispersion results (TSI = 0.15). The accelerated weathering test (fog/UV-vis) seemed to have an important impact on the physicochemical properties of the different coatings, and was most pronounced on their thermal properties and curing degree. Before the weathering test, the nature of the nanoparticle played an important role in the water permeability and adherence results, since less hydrophilic composites, i.e., SMNC-S10-1, and AC-S10-1, showed the lowest weight gain and best adherence in the water permeability and pull-off tests, respectively. Additionally, the latter composite presented the best abrasion resistance. After the accelerated weathering test, all composites presented a lower weight gain in comparison to the neat epoxy resin, especially ZnO-S10-1. The highest Tg values obtained with AC-S10-1 and SiO_2_-S10-1 composites demonstrated the best adherence and abrasion resistance results.

To our knowledge, no other study has reported the incorporation of nanoparticles into epoxy resins and their abrasion and adherence properties after weathering tests. This research opens a new starting point for long-term coatings in concrete for future industrial applications.

## Figures and Tables

**Figure 1 nanomaterials-11-00869-f001:**
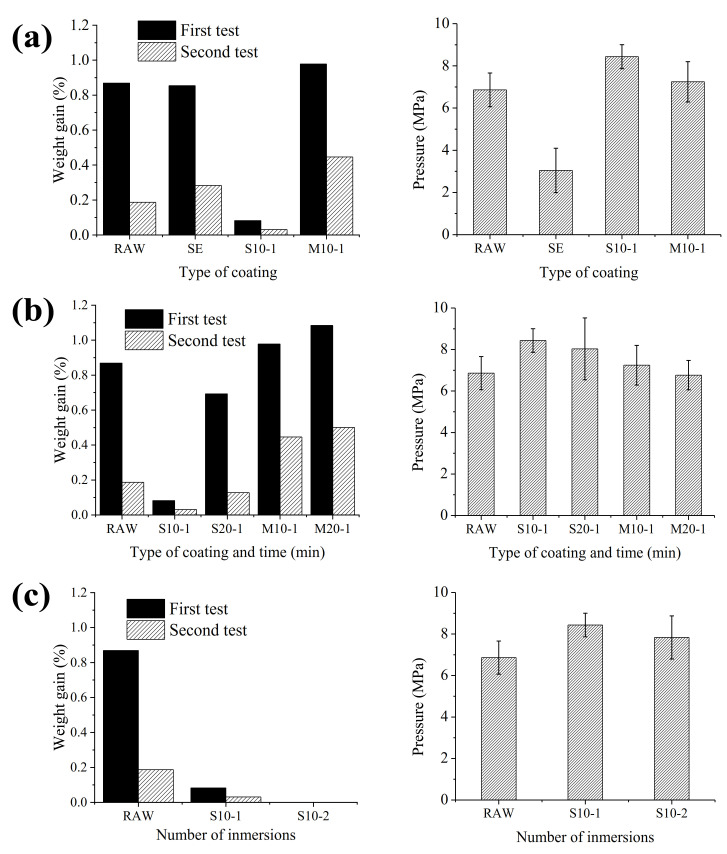
Water permeability and pull off results in the study of the effect of (**a**) the type of coating, (**b**) the immersion time in the dip-coating methodology, and (**c**) the number of the dip-coating immersions.

**Figure 2 nanomaterials-11-00869-f002:**
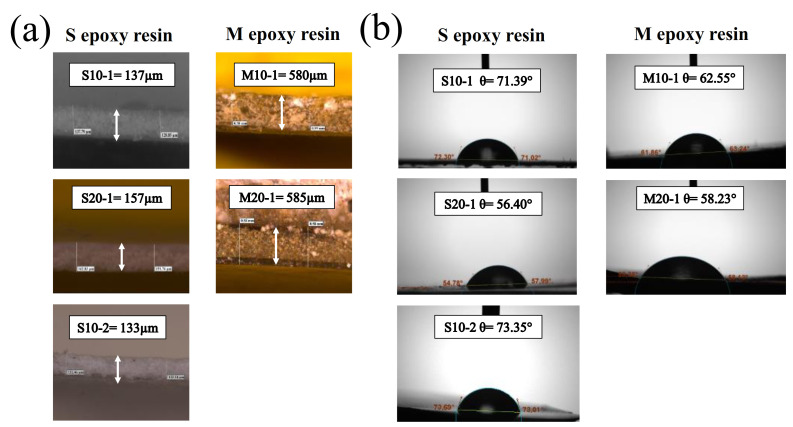
Thickness of the different S and M coatings (**a**). Contact angle analyses of the different S and M coatings (**b**).

**Figure 3 nanomaterials-11-00869-f003:**
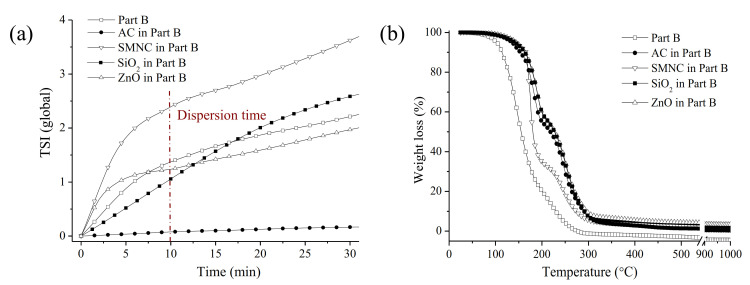
Stability analyses of the different NP incorporation into Part B of the S resin (**a**). TGA analyses of NPs-Part B composites (**b**).

**Figure 4 nanomaterials-11-00869-f004:**
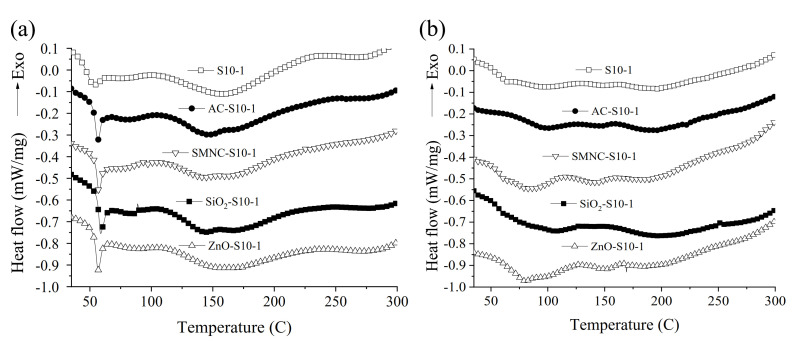
DSC results of the different composites before (**a**) and after (**b**) the weathering test.

**Figure 5 nanomaterials-11-00869-f005:**
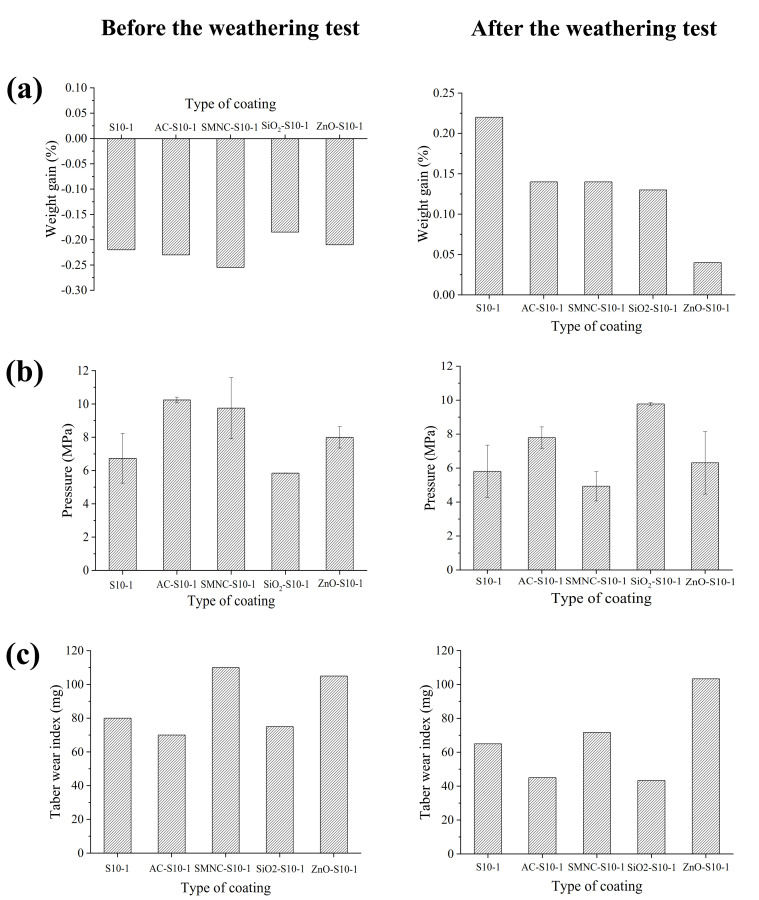
Results of water permeability (**a**), pull-off test (**b**) and abrasion resistance test (**c**) before and after accelerated weathering test of the different coated concretes.

**Table 1 nanomaterials-11-00869-t001:** Characterization results of the different coatings before (**a**) and after (**b**) the accelerated weathering test.

**(a)**	**NPs-S10-1 before the Weathering Test**					
**Technique**	**Parameter**	**S-10**	**AC-S10-1**	**SMNC-S10-1**	**SiO_2_-S10-1**	**ZnO-S10-1**
Contact Angle	θ (°)	71.6	77.8	88.7	63.8	59.7
Optical Loop	Thickness (µm)	137.8	93.2	85.2	182.7	164.4
DSC-N_2_	Tg_1_ (°C)	52.7	56.9	57.6	58.6	56.8
TGA-N_2_	T_5%_ (°C)	160.3	146.8	152.5	163.1	166.1
	T_40%_ (°C)	400.1	407.2	419.7	405.3	404.2
	Weight Loss_stage1_ (%)	10.4	11.9	10.4	10.4	10.4
	Weight Loss_stage2_ (%)	53.1	47.4	45.9	48.4	49.2
	Residue (%)	21.6	24.5	23.9	24.3	24.3
TGA-Air	T_5%_ (°C)	138.1	134.8	142.3	147.2	154.3
	T_40%_ (°C)	391.7	382.8	389.1	389.2	385.1
	Weight Loss_stage1_ (%)	12.2	14.6	12.8	12.5	12.5
	Weight Loss_stage2_ (%)	41.7	40.6	41.4	42.6	41.7
	Residue (%)	20.6	16.3	15.1	23.1	22.6
**(b)**	**NPs-S10-1 after the Weathering Test**					
**Technique**	**Parameter**	**S-10**	**AC-S10-1**	**SMNC-S10-1**	**SiO_2_-S10-1**	**ZnO-S10-1**
Contact Angle	θ (°)	55.7	66.1	71.4	58.8	77.9
Optical Loop	Thickness (µm)	127.4	112.7	113.5	129.5	84.1
DSC-N_2_	Tg_1_ (°C)	63.8	98.5	80.9	104.9	79.7
TGA-N_2_	T_5%_ (°C)	201.7	195.2	191.3	212.7	206.0
	T_40%_ (°C)	412.1	414.5	417.5	411.2	410.7
	Weight Loss_stage1_ (%)	8.4	7.3	7.3	8.4	7.3
	Weight Loss_stage2_ (%)	50.1	46.6	46.6	48.1	46.7
	Residue (%)	23.5	26.6	26.2	26.5	28.5
TGA-Air	T_5%_ (°C)	175.5	177.1	154.3	173.1	171.3
	T_40%_ (°C)	401.5	404.8	400.2	392.5	398.2
	Weight Loss_stage1_ (%)	8.7	8.7	9.9	10.7	8.7
	Weight Loss_stage2_ (%)	41.2	37.9	39.3	44.7	41.2
	Residue (%)	23.3	26.5	23.2	22.1	23.1

## Data Availability

Data is available on reasonable request from the corresponding author.

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
