# Peer review of "Long-Term Performance of Nanomodified Coated Concrete Structures under Hostile Marine Climate Conditions"

_nanomaterials, 2021, doi:10.3390/nano11040869_

Round 1

Reviewer 1 Report

Authors submitted their manuscript to Nanomaterials, however, according to  my opinion, it was not a good choice. Although the particle “nano” may be found in many places in the manuscript, even in the tittle, in fact authors do not discuss any “nano” aspects of the problem they would like to solve. Initial part (fig. 1) does not refer to “nano” aspects at all. I would like to remind authors that material can be classified as nano matter if at least one dimension of its particles is below 100 nm. Is that condition fulfilled in the case of the considered manuscript? I do not know because no characteristics of the materials are given. Anyway this nano aspect does not seem to play any role in the authors’ work. According to my opinion this manuscript should be submitted to a journal which subject is coatings.

Authors placed much of the results in the supporting information (SI). There are only 5 figures in the manuscript, whereas in SI there are as many as 9 figures.  In many places of the manuscript authors discuss the results, referring to the figures in the SI. Unfortunately I was not able to access those material. Supporting information should be accessible to the referees or authors should not place there any important information.

The manuscript is written in the inconsistent manner. The test in figure 1a is called “Water absorption test” and the ordinate label is “Adsorbed water (%)”, in figure 5 the same (?) test is called “water permeability” (in caption) and the ordinate label is “Weight gain (%)”. By the way: what is the reason that this gain is negative? In figure 1 there are presented two tests: First test and Second test. Is the second test the repetition? If so, what is the reason for such large difference?

In figure  1: the sample S20-1 absorb many times more water than the sample S10-1. The difference in treatment is, that sample S20-1 was kept immersed in the resin 20 seconds, the sample S10-1 ten seconds. What was the reason, that prolongation of the contact with the resin increased the susceptibility to adsorption so much? It is strange and authors should explain that phenomenon. There are more such unexpected results in the reviewed manuscript.

Authors presented many interesting results, however there are many inconsistences in their manuscript, so I suggest  reconsideration by the authors and re-submission, preferably rater to Coatings, not to Nanomaterials.

Author Response

A document is attached 

Reviewer 2 Report

In the presented manuscript, the authors present the results of research on the influence of the method of applying epoxy-based protective coatings to concrete substrates and the type of filler used for their modification on their resistance to weather conditions. Although the undertaken research topic seems important especially from the utility point of view, in my opinion, the method of presenting the obtained results does not allow the publication of the submitted manuscript in its current form.

  • The authors did not provide a supplementary materials file which, according to the manuscript, contains the information necessary for its proper evaluation.
  • The description of the starting materials used (epoxy resins and fillers) is insufficient, which makes it impossible to repeat the experiments. The authors did not provide specific trade names of the resins and hardeners used. There are also no important parameters of the fillers used given (their specific surface area, particle size, and distribution, density, etc.) that have a significant impact on the properties of the obtained composites and their assessment.
  • The description of the analytical techniques used does not provide information on the type of adhesive and the dolly diameter used in the pull-off tests.
  • The authors suggest the dip-coating procedurÄ™ as the most effective method of impregnation of concrete elements. This statement is perhaps correct in the case of laboratory tests (although the authors did not provide direct experimental evidence for it), however, it is difficult to imagine the use of this technique for securing large concrete constructions. Moreover, the authors investigated the effect of the number of dips of concrete samples on the quality and properties of the obtained protective coatings, however, the procedure did not indicate the time after which the samples were immersed again after the first immersion. Before or after the first coat was cured?
  • Optimization of the method of mixing the filler with the resin was performed only for the hardener/filler system, while the degree of dispersion of the filler in the resin mixed with the hardener was not determined.
  • The interpretation of the DSC analysis results seems to be incorrect, mainly because the analyzes were performed in the wrong temperature range and only for the first heating run. In my opinion, the beginning of the measurement should be shifted towards lower temperatures (e.g. down to -20C) and should not exceed the temperature of the beginning of sample decomposition. The Exo direction is not shown in the charts of the DSC curves.
  • In lines 499-501 the authors write: "However, the highest Tg values obtained with AC-S10-1 and SiO2-S10-1 composites, demonstrated the high affinity between these NPs and the epoxy resin, promoting the best adherence and abrasion resistance results." This statement is inconsistent with the results shown in Figure 5 for samples after the weathering test.

Author Response

A document is attached

Reviewer 3 Report

The research topic, namely nanomodified coatings for concrete structure, is interesting, especially for civil engineering applications.

The scientific approach is based on exprimental tests, especially weathering test.

The main results are significant and demonstrate that the nanomodified coatings are effective.

Author Response

A document is attached

Round 2

Reviewer 1 Report

Authors improved significantly the manuscript and now I have only one comment. In many places in the manuscript, especially in the table in page 14, authors give the results of the measurements with 4 or 5 significant digits. The number of significant digits should reflect the accuracy of the measurement.  For example writing the number as 173.00 (taken from the table) means that the accuracy of the measurement was +0.01. I do not believe that author make all measurements with such accuracy. Note, that even if the manufacturer of the instrument guarantees such precision, it does not mean the measurements were done with such accuracy, because the measured quantity may depends on some uncontrolled parameters. That may be checked by repeating the measurement several times and calculating the standard deviation std. The accuracy of the measurement is not better than the calculated std is.

Author Response

A document is attached.

Reviewer 2 Report

In my opinion, the information presented in Table S1 regarding the resins used is still insufficient. Without data describing the tested materials (resins), it is difficult to relate the obtained results to those obtained by other research groups. For this reason, I consider the Significance of Content to be low. 

Author Response

A document is attached.
